# Old Player-New Tricks: Non Angiogenic Effects of the VEGF/VEGFR Pathway in Cancer

**DOI:** 10.3390/cancers12113145

**Published:** 2020-10-27

**Authors:** Panagiotis Ntellas, Leonidas Mavroeidis, Stefania Gkoura, Ioanna Gazouli, Anna-Lea Amylidi, Alexandra Papadaki, George Zarkavelis, Davide Mauri, Georgia Karpathiou, Evangelos Kolettas, Anna Batistatou, George Pentheroudakis

**Affiliations:** 1Department of Medical Oncology, University Hospital of Ioannina, 45500 Ioannina, Greece; ntellasp@gmail.com (P.N.); leo.mavroidis@gmail.com (L.M.); s.gkoura@uoi.gr (S.G.); ioannagazouli@gmail.com (I.G.); annalea.ami@gmail.com (A.-L.A.); alexpapadaki@yahoo.gr (A.P.); g.zarkavelis@uoi.gr (G.Z.); dmauri@uoi.gr (D.M.); 2Society for Study of Clonal Heterogeneity of Neoplasia (EMEKEN), 45445 Ioannina, Greece; 3Department of Pathology, University Hospital of St-Etienne, 42055 Saint Etienne, France; georgia.karpathiou@chu-st-etienne.fr; 4Laboratory of Biology, School of Medicine, Faculty of Health Sciences, University of Ioannina, 45110 Ioannina, Greece; ekoletas@uoi.gr; 5Biomedical Research Division, Institute of Molecular Biology & Biotechnology, Foundation for Research & Technology, 45115 Ioannina, Greece; 6Department of Pathology, University Hospital of Ioannina, 45500 Ioannina, Greece; abatista@uoi.gr

**Keywords:** angiogenesis, VEGF, VEGFR, anti-angiogenesis, anti-angiogenic agents, tumor progression, immunosuppression, immunotherapy, immune-checkpoint inhibitors, combination therapy

## Abstract

**Simple Summary:**

Although VEGF-A is well characterized as the principal player of cancer angiogenesis, new data on the interplay with other components of the tumor microenvironment emerge. Here we review the effect of VEGF-A on cancer cells and immune cells as well as investigative and established combinational therapies of anti-angiogenic agents with immune checkpoint inhibitors. We thus elaborate the scientific rationale behind the development of these novel combinational approaches.

**Abstract:**

Angiogenesis has long been considered to facilitate and sustain cancer growth, making the introduction of anti-angiogenic agents that disrupt the vascular endothelial growth factor/receptor (VEGF/VEGFR) pathway an important milestone at the beginning of the 21st century. Originally research on VEGF signaling focused on its survival and mitogenic effects towards endothelial cells, with moderate so far success of anti-angiogenic therapy. However, VEGF can have multiple effects on additional cell types including immune and tumor cells, by directly influencing and promoting tumor cell survival, proliferation and invasion and contributing to an immunosuppressive microenvironment. In this review, we summarize the effects of the VEGF/VEGFR pathway on non-endothelial cells and the resulting implications of anti-angiogenic agents that include direct inhibition of tumor cell growth and immunostimulatory functions. Finally, we present how previously unappreciated studies on VEGF biology, that have demonstrated immunomodulatory properties and tumor regression by disrupting the VEGF/VEGFR pathway, now provide the scientific basis for new combinational treatments of immunotherapy with anti-angiogenic agents.

## 1. Introduction

Over the last two decades, since the Cancer Genome Atlas Project (TCGA, https://www.cancer.gov/about-nci/organization/ccg/research/structural-genomics/tcga) started, our understanding of cancer biology has grown exponentially and has paved the way for new exciting treatment modalities. Indeed, the advancements in cancer research have brought the development of new anticancer drugs, radiation therapy devices, surgical techniques, diagnostic methods, prognostic and predictive biomarkers and prepared the ground for precision oncology, all of which have contributed to the survival and quality of life of cancer patients [1]. Despite these advances, the burden of cancer in developed societies remains high; malignancies are still the 2nd leading cause of death with 599,108 cancer-related deaths in the US alone, in 2017 [2]. Amongst the most important therapeutics that have driven cancer treatment the last two decades would be the introduction of anti-angiogenic therapy in 2004 and the emergence of immune-checkpoint inhibitors in 2011.

Already from the 1940s [3,4], the release of “blood-vessel growth-stimulating factors’’ that would have that the ability to induce new vessel growth, was hypothesized to confer a growth advantage to tumors [5]. Following an observation that rapidly growing tumors were heavily vascularized, Folkman et al. [6,7,8] were the first to isolate a factor from animal tumors that could stimulate angiogenesis and suggested, almost 5 decades ago, that ‘‘anti-angiogenesis’’ could be a strategy to treat cancer [5]. It was not until 1993 that the use of antibodies against the vascular endothelial growth factor (VEGF) in immune-deficient mice successfully suppressed tumor growth [5,9]. The murine anti-VEGF antibody used in the preclinical tumor models was humanized [10] and the recombinant antibody later known as bevacizumab was granted Federal Drug Administration (FDA) approval in 2004 for metastatic colorectal cancer [5]. Bevacizumab is an antibody against vascular endothelial growth factor A (VEGF-A) and following its success several other strategies to inhibit the vascular endothelial growth factor/receptor (VEGF/VEGFR) pathway were devised, namely: receptor tyrosine kinase inhibitors, anti-VEGFR2 antibodies and a VEGF trap (i.e., a soluble VEGF receptor). Bevacizumab also gained expanded approval for several different malignancies including non-small cell lung carcinoma (NSCLC) ovarian and renal cell carcinoma (RCC) [5,11,12]. However, anti-angiogenic therapy in general has managed to offer only modest survival benefits before resistance develops [8], and bevacizumab’s benefit was only evident when combined with cytotoxic chemotherapy [13].

No doubt, VEGF is established as an indispensable regulating factor of angiogenesis, contributing to vascular homeostasis, and when dysregulated to disease, with proof of principle anti-VEGF therapy studies demonstrating anti-tumor efficacy by inducing regression of blood vessels [5,13,14]. Despite the moderate so far success of anti-angiogenic therapy, and while VEGF mainly targets endothelial cells, it has been demonstrated that this factor has multiple effects on additional cell types, including immune and tumor cells [13,15,16]; thus, implicating VEGF in diverse molecular pathogenic processes that drive tumor progression, unrelated to the stimulation of angiogenesis [5]. This review focusses on the effects of the VEGF/VEGFR pathway on non-endothelial cells and the resulting unconventional implications of anti- angiogenic agents, other than pruning of new blood vessels.

## 2. The VEGF/VEGFR Pathway

The process of vessel formation, either through vasculogenesis or angiogenesis, is regulated by numerous receptors that are predominantly expressed on endothelial cells [17,18]. VEGFRs are the most known and well-studied family of endothelial specific receptors, but others include the Tie and Ephrin (Eph) receptor families. While ephrin receptors are mainly involved in arterial-venous specification, VEGF receptors regulate endothelial differentiation and initiation of angiogenesis or vasculogenesis, and Tie receptors control later stages of vessel formation such as stabilization of the endothelial sprout [17,19,20]. In addition, the Notch signaling pathway is critical for the coordination of the multistep process of angiogenesis through specification of the tip or stalk cell phenotype [21].

The family of the VEGF receptors is comprised of the VEGFR1, VEGFR2 and VEGFR3 receptors, which although show similar overall structural organization, still they display differences in their mode of activation, signaling and biological effects [22]. VEGFRs contain multiple tyrosine residues in their cytoplasmic domain and possess intrinsic tyrosine kinase activity [17]. Binding of VEGFs to VEGFRs induces receptor homo- or hetero-dimerization, leading to autophosphorylation of the tyrosine residues. Phosphorylated tyrosines of the VEGFRs’ intracellular domains act as binding sites for adaptor molecules, activating downstream signaling pathways [22]. Apart from VEGF-triggered signaling, VGFRs can also undergo non-VEGF-dependent activation, and this VEGFR non-canonical signaling can be induced by binding of non-VEGF ligands or shear-stress-activated cytoplasmic SRC tyrosine kinases [22]. All three VEGFR receptors can trigger cell survival and proliferation, similar to other growth factor receptors; however, they can also provide specific signals that mediate endothelial cell-specific functions for vessel formation [17]. VEGFR1 and 2 are primarily expressed on endothelial cells, while expression of VGFR3 is mainly restricted on lymphatic cells, although VGFR3 is also involved at the first stages of vessel formation in the embryo [17]. VEGFR1 is a high affinity tyrosine kinase receptor for VEGF-A, however, it displays weak ligand-dependent autophosphorylation [5,17,23] and has been suggested to act as a decoy receptor for VEGF-A, preventing it from binding with VEGFR2 [5,17]. VEGFR2 displays weaker VEGF-A binding affinity; however, VEGFR2 has been established as the main signaling receptor for VEGF-A promoting vascular endothelial cell mitogenesis, permeability and cell migration [5,17,24,25].

VEGFR signaling is also modulated by different co-receptors. Specifically, VEGFs as well as VEGFRs bind to co-receptors such as heparan sulphate proteoglycans (HSPGs) and neuropilins (NRPs), such as NRP1 and NRP2 [26]. Oddly, NRP receptors lack intrinsic catalytic activity, but can enhance endothelial cell activity, in response to VEGF signaling [17]. These interactions can influence VEGFR mediated responses, for example, by affecting the half-life of the receptor complex or VEGFR phosphorylation [26,27].

The family of vascular endothelial growth factors (VEGFs) includes VEGF-A, which is the first member described and is usually simply referred to as VEGF, as well as VEGF-B, VEGF-C, VEGF-D and placenta growth factor (PLGF) [15,22]. These structurally-related dimeric proteins are broadly expressed and play a central role in vascular homeostasis by binding to specific receptor tyrosine kinases, most notably VEGFRs [22]. VEGFs also display high affinity to VEGF co-receptors, namely NRP receptors and HSPGs [22]. VEGF-A binds both to VEGFR1 and 2, while VEGF-B and PLGF are selective for VEGFR1 [5,17,28]. VEGF-C and D are primarily ligands of VEGFR3 implicating them in the regulation of lymphangiogenesis, but they can also bind to VEGFR2 after being proteolytically processed [5,17,28,29]. Binding of VEGF-A to VEGFR2 is considered the main signaling event triggering angiogenesis [15], as highlighted by the embryonic lethality of mice lacking expression of either VEGF-A or VEGFR2 [30,31]; justifiably, most attention has been focused on VEGF-A [5]. The VEGF-A gene contains eight exons, that by alternative splicing give rise to different isoforms [32,33,34,35]. The VEGFxxx variants, where xxx denotes the number of aminoacids in VEGF-A protein, come up by alternative splicing of the exons 5–7 while alternative splicing of exon 8 give rise to either the VEGFxxxa or VEGFxxxb isoforms with pro-angiogenic and anti-angiogenic properties respectively [36,37] and potential predictive role on anti-VEGF treatment [35]. Indicative isoforms are the VEGF_121 a_, VEGF_121 b_, VEGF_165 a_, VEGF_165 b_, VEGF_189 a_, VEGF_189 b_, VEGF_206 a_ and VEGF_206 b_ with different profiles on activity and bioavailability with VEGF-A_165 a_ being the most extensively investigated [32,33,34,35]. Neo-angiogenesis and vascular permeability constitute the main pathogenic effects mediated by VEGF-A [5].

Although VEGF-A is the principal player that initiates sprouting angiogenesis, new vessel formation would not be possible without the joint action of the Notch signaling pathway [38]. VEGF-A promotes migration of endothelial cells towards a gradient of angiogenic factors in the tumor microenvironment. The leading role is taken by the tip cell that senses the external signals through extension of filopodia and increased expression of VEGFR2 [39]. However VEGF-A also induces the expression of the Notch ligand Delta-Like 4 (DLL4) in the tip cell which sequentially activates Notch signaling in the adjacent endothelial cells [40,41]. The latter results in a decrease of VEGFR2 [42] and increase of VEGFR1 expression in the neighbor cells and acquisition of a stalk cell phenotype [43]. Therefore, it is established an angiogenic front by tip cells that guide the new sprout and a thread of stalk cells that constitute the scaffold of the new vessel [21].

## 3. Autocrine Effects on Cancer Cells

The release of pro-angiogenic mediators, and of VEGF in particular, has been described for various solid tumors and hematologic malignancies [32,44,45,46,47,48,49,50,51]. Initially research on VEGF signaling focused on its survival and mitogenic effects towards endothelial cells [52,53], with stimulation of angiogenesis being considered the primary mechanism of VEGF mediated cancer progression and metastasis; not surprisingly, since VEGFRs were traditionally regarded to be restricted on the vascular endothelium [54]. However, over the years, expression of VEGFRs has been described on several types of non-endothelial cells, including cancer cells [55,56,57]. It is now conceivable that tumor-derived VEGF not only provides paracrine signaling for endothelial cells, but may also directly stimulate tumor growth in an autocrine manner [54,58]. Therefore, VEGF-blockade may act on multiple levels: antiangiogenic effects on the tumor vasculature and antineoplastic effects on the tumor cell population [16]. While these antineoplastic effects can be easier assessed in tumor cell lines, further investigation is warranted for their clinical relevance (Figure 1 and Table 1).

### 3.1. Melanoma

The hypothesis that the VEGF/VEGFR pathway would play an autocrine role in tumor progression began from the observation that many malignant cells co-express VEGF and its receptors [59,93]. In 1995, VGEFR2 was detected on three melanoma cell lines (i.e., MeWo, A375-metastatic, A375-wt), that were also known to co-express VEGF; intriguingly exogenous administration of VEGF increased the proliferation of A375 M melanoma cells in vitro [59]. Another study confirmed the expression of VEGFR2, VEGFR1, NRP1 (Neuropilin1), NRP2 (Neuropilin2) and production of VEGF_121_, VEGF_165_, VEGF_189_ and PLGF in melanoma cell lines derived from primary or metastatic tumors (i.e., GR-Mel, ST-Mel, SN-Mel, PR-Mel, CN-Mel, TVMBO, SK-Mel-28, WM115, WM266–4, 13443-Mel, PDMel, PNP-Mel, PNM-Mel, LCP-Mel, LCM-Mel, GL-Mel, M14, LB-24, 397-Mel). Exposure of the VEGFR-expressing melanoma cells to VEGF_165_ and PLGF-1 resulted on a proliferative response, while M14 cells lacking VEGFR1 and 2 were unresponsive [93]. In addition, stimulation of melanoma cells was inhibited by neutralizing anti-VEGF antibodies and was completely abolished with anti-PLGF antibodies, confirming the specificity of the response [93]. Likewise, NRP1 was reported to mitigate migration of melanoma cells (i.e., M14, GR-Mel) though VEGF-A-induced activation of VEGFR2 [60], or independently in response to PLGF, even in the absence of its high affinity receptor, VEFGR1 [94].

### 3.2. Pancreatic Cancer

Expression of VEGFRs has also been demonstrated in pancreatic cancer [44,61]. Analysis of pancreatic cancer tissues revealed concomitant over-expression of VEGF and of its high affinity receptors in 33% of pancreatic cancer patients [44]. VEGFR2 expression was observed in three pancreatic cell lines (AsPC-1, Capan-1 and MIAPaCa-2) and VEGFR1 mRNA was detected in four pancreatic cancer cell lines (AsPC-1, Capan-1, T3 M4 and PANC-1). Furthermore, radiolabeled VEGF was detected to bound to Capan-1 pancreatic cancer cells, which also exhibited enhanced MAPK activation and growth upon VEGF stimulation, demonstrating evidence of a VEGF/VEGFR2 autocrine signaling [44]. In another study, VEGFR1, but not VEGFR2, appeared to be ubiquitously expressed in pancreatic carcinoma cell lines (i.e., AsPC-1, BxPC3, CFPAC, HPAF2, MiaPaCa2, Panc-1, HS7665, Panc-48, L3.6 pl, FG) with concomitant expression of its ligands (i.e., VEGF-A and VEGF-B) [61]. Further analysis of the L3.6 p1 and Panc-1 cells revealed that both VEGF-A and VEGF-B induced ERK1/2 phosphorylation mediated through VEGFR1, as none of the cell lines examined were found to express VEGFR2. Use of a neutralizing antibody to VEGFR1 confirmed that the signaling was VEGFR1-dependent [61]. Further analysis demonstrated that migration and invasion of pancreatic cancer cells was promoted upon VEGFR1 stimulation, on the contrary, no effect was observed on cell proliferation [61]. NRP1 was also reported to contribute to pancreatic cancer aggressiveness by promoting transforming growth factor beta 1 (TGFβ1)-induced fibrosis and endothelial-to-mesenchymal cell transition, a process that serves as an important source of fibroblasts [62].

### 3.3. Lung Cancer

VEGFR-2, VEGFR-3, VEGF-A and VEGF-C have also been detected in small cell lung cancer (SCLC) cell lines (i.e., NCI-H82, H209, H510, H526 and H660) [64]. Stimulation by VEGF-A and VEGF-D induced phosphorylation of VEGFR2 and VEGFR3 respectively, as well as increased ERK1/2 phosphorylation and proliferation of these cells [64]. VEGF_165_, as well as VEGFR1, VEGFR2, NRP1 and NRP2 were detected in several NSCLC cell lines examined (i.e., H460, H647, A549, SKMES1). VEGF-A was found to stimulate the proliferation of NRP1-expressing cells in the presence of VEGFR2 [63]. In addition, phosphorylation of the PI3K-mediator Akt and, to a lesser extent, of the MAPK’s signaling proteins ERK1/2, was demonstrated in A549 and SKMES1 NSCLC cell lines treated with VEGF-A [63].

### 3.4. Gastrointestinal Cancer

Among the tumors of epithelial origin, expression of VEGFRs has been observed on those arising from the colon [66]. VEGFR1 has been detected in a series of colon cancer cell lines (i.e., HT29, SW480, SW620, ATCC, KM12 L4, KM12 SMLM2, GEO, RKO) [67], with evidence suggesting that the receptor is involved in processes that promote tumor progression and metastasis [66,67]. Likewise, upregulation of both VEGF and VEGFR1, but not of VEGFR2, has been detected in LIM1863 colon cancer cells undergoing an epithelial-to-mesenchymal cell transition (EMT). Importantly, VEGF/VEGFR1 autocrine interaction appeared to be necessary for the survival of the LIM1863 colon carcinoma cells after the induction of EMT [68,69]. Furthermore, RNAi-mediated depletion of VEGF decreased cell survival and enhanced sensitivity to chemotherapy of colorectal cancer cells (i.e., HCT116, SW480, HT29, HCP-1) by disrupting AKT and ERK1/2 signaling; notably, ribonucleic acid interference (RNAi)-mediated depletion of VEGFR1 replicated the effects of VEGF depletion on phospho-AKT and phospho-ERK1/2 levels [65]. Moreover, VEGF-A and VEGFR1/2 are widely expressed in gastric carcinoma cells (i.e., RF-1, RF-48, AGS-1, NCI-N87, NCI-SNU-1, NCI-SNU-5, NCI-SNU-16, KATO-III). Tumor growth was found to be enhanced in VEGFR2-positive cells after VEGF-A stimulation, but not in gastric adenocarcinoma cells expressing only VEGFR1 [57,70].

### 3.5. Prostate Cancer

VEGFR1 and 2 expression, with concomitant VEGF production has been observed in prostate cancer cells (i.e., LNCaP, PC3, DU145) [71,95], as well as in prostate cancer tissue specimens [71,72]. Malignant cells particularly displayed greater receptor expression compared to normal basal prostate cells [71]. Furthermore, the LNCaP prostate cancer cell line demonstrated 50% enhanced proliferation in the presence of VEGF_165_, an effect that was abolished by a neutralizing antibody to VEGFR2, suggesting that the survival signals from VEGF are mediated specifically via VEGFR2 [71].

### 3.6. Gliomas

While several studies report that World Health Organization (WHO) grade IV gliomas (i.e., glioblastomas) secrete high levels of VEGFs, expression of VEGFRs on grade IV glioma cells (i.e., U118, U138, U343, U87) and primary glioblastoma cell lines has been mostly found to be weak [73,96,97]. Despite a low VEGFR2 and no VEGFR1 expression, drugs targeting the VEGF pathway demonstrated biological effects on cell proliferation, morphology and metabolism in the U87 glioma grade IV cell line [97]. Furthermore, higher VEGFR1 or VEGFR2 mRNA expression levels, in grade II, III and IV glioma patients, have been corelated with higher tumor grade and worse prognosis [73]. Additionally, activation of MAPK/ERK, PI3K/Akt and PLC/PKC pathways was found to be induced by VEGF through VEGFR2 and VEGFR1 signaling in a panel of grade III/IV glioma cell lines [73,74]. Likewise, in vivo studies indicate that VEGFR1 and VEGFR2 signaling support survival of orthotopic glioma bearing mice [73]. Furthermore, proliferation of glioblastoma stem-like cells was shown to be stimulated via VEGFR2 by exogenous VEGF in a dose-dependent matter, but not via VEGFR1. On the contrary VEGFR1 seemed to have a negative feedback effect on VEGFR2 when cells were exposed to higher concentrations of VEGF [75].

### 3.7. Breast Cancer

Production of VEGF and expression of VEGFR1 and 2 has been described in breast cancer tissues [98] and in several primary breast cancer cell lines [32], with in vitro studies demonstrating that ^125^I-labeled VEGF can bind to T-47 D cells and by doing so to induce activation of the MAPK/ERK and PI3K/Akt pathways [76]. In addition, data from a transgenic mouse model with human VEGF_165_ targeted to mammary epithelial cells, indicated that VEGF-A contributes to mammary tumor growth, not only through increased neovascularization, but also by stimulating the proliferation of tumor cells in an autocrine manner, and by inhibiting their apoptosis [77]. Specifically, expression of VEGFR1 has been reported in a panel of breast cancer cell lines (i.e., DU4475, MCF-7, T-47 D, SK-BR-3, MDA-MB-157, MDA-MB-175, MDA-MB-231, MDA-MB-435, MDA-MB-468, AU565, BT-474, BT-483, HCC38, UACC-812, ZR-75–1), followed by the observation that tumor cell growth is supported by selective VEGFR1 signaling and it is mediated by downstream activation of MAPK/ERK and PI3K/Akt pathways [99]. VEGFR2 expression has also been established in breast cancer specimens [98,100], along with concomitant VEGF expression [98,101]. Moreover, in a series of 142 invasive breast carcinomas, 64.5% of them tested positive for VEGFR2 expression and were also associated with the expression of Ki67 and topoisomerase-IIa proliferation indexes suggesting that VEGF may act as a growth factor via VEGFR2 in these cancer cells [102]. VEGFR2 phosphorylation in several breast cancer cell lines (i.e., MDA-MB-468, T47 d, MCF-7, HBL-100 and in a primary breast cancer culture) was enhanced by VEGF-A stimulation leading to activation of ERK1/2 and Akt pathways, indicating that the VEGFR/VEGF-A pathway might play crucial role in the regulation of survival and proliferation of breast cancer cells [78]. VEGF-A was also reported to drive self-renewal of breast and lung cancer stem cells by stimulating the VEGFR2/Stat3 signaling and inducing *Myc* and *Sox2* expression [84]. Likewise, the VEGF-A/NRP1 axis was suggested to confer cancer stem cell traits in breast cancer cells (i.e., MCF-7, MDA-MB-231) by activating the Wnt/β-catenin pathway [85]. In addition, the VEGF-A/NRP1 axis was associated with breast cancer progression by enhancing the EMT process and NF-κB (nuclear factor kappa-light-chain-enhancer of activated B cells) and β-catenin signaling [82], with further evidence to support that neuropilin might also protect MDA-MB-231 breast cancer cells from apoptosis by autocrine stimulation of the PI3K-pathway in response to VEGF_165_ [79]. Likewise, NRP1 gene silencing was reported to suppress the proliferation, promote apoptosis and increase the sensitivity of breast cancer cells (i.e., MCF-7, SK-BR-3) to chemotherapy [83].

### 3.8. Hematologic Malignancies

VEGF expression has been observed in hematologic malignancies [47], with evidence to suggest that VEGF triggers growth, survival and migration of leukemia and multiple myeloma (ΜΜ) cells [88,103,104]. The VEGF/VEGFR-induced activation of intracellular tyrosine kinase cascades in MM has been described since 2001 [89]. Specifically, the VEGF/VEGFR-triggered MAPK/ERK pathway was found to mediate MM cell proliferation, while the PI3 k/PKC–dependent cascade was associated with migration and the myeloid cell leukemia 1 (McL1)/survivin with survival [88,89]. VEGFR1 was found to be more widely expressed in MM cells compared to VEGFR2 [88,90,91]. Likewise, stromal derived VEGF-A was shown to induce VEGFR1-dependent proliferation of primary MM cells, while in vitro inhibition of MM cell lines (i.e., RPMI 8226, U266, ARP1, ARK) by bevacizumab resulted in a reduction of proliferation [92]. Recently, the junctional adhesion molecule-A (JAM-A) has emerged as a crucial mediator between MM plasma and medullary endothelial cells, and has been associated with poor prognosis of MM patients due to its role in invasion and metastasis [105,106]; while limited so far, evidence suggests that JAM-A could also interfere with the VEGF/VEGFR pathway [107]. Similarly, VEGF induced phosphorylation of VEGFR2 expressing leukemia cells (i.e., HL-60, HEL and primary leukemia cell lines), resulting in increased proliferation [51]. VEGF may also facilitate survival of leukemia cells by up-regulation of heat shock protein 90 (Hsp90), which deactivates significant pro-apoptotic molecules [89], and was also corelated with increased expression of the anti-apoptotic MCL-1 gene in B- chronic lymphocytic leukemia patients [108].

### 3.9. Other

Several other reports on a variety of additional malignancies suggest that VEGF may act in an autocrine loop fashion in cancer cells. For example, in head and neck (H&N) cancer, where VEGFR2 was detected in 109 H&N squamous cell tumors, with evidence to suggest that the receptor might regulate proliferation and invasion of H&N cancer cells (i.e., Hep2) [86]. VEGF-A, VEGFR1 and 2 expression is also present in bladder cancer, with VEGFR2 found particularly prominent in muscle invasive bladder cancer specimens [87]. Additionally, several bladder cancer cell lines exhibit VEGFR expression, with T24 cells displaying enhanced survival and proliferation, mediated by VEGFR2 in response to VEGF signaling [45]. Furthermore, expression of both VEGFR1 and VEGFR2 has been detected on multiple rhabdomyosarcoma cell lines (i.e., RH4, RH6, RH18, RH28, RD), with the VEGFR1-positive cell lines demonstrating increased proliferation upon VEGF_165_ stimulation, while proliferation was halted after applying a blocking antibody against VEGFR1 [52]. VEGF-A has also been detected in the ovaries, both in normal and cancer tissues, and found to be secreted in malignant ascites, with epithelial cancer cells being identified as the source of VEGF-A [46,48,49]. VEGFR2 displayed a more prominent expression in ovarian cancer specimens and cell lines (A2774, SKOV3 ip1, HeyA8) as compared to normal ovarian samples where little to none VEGFR2 is detected [58]. VEGFR2 was also found phosphorylated in ovarian cancer cells and has been correlated with their proliferation and survival [58]. VEGFR1 on the contrary was largely absent [58].

### 3.10. VEGF Signaling on Cancer Cells: Stimulation of Survival and Migration

The signaling pathways activated by VEGF have been well characterized in endothelial cells [26,44]. VEGF-induced phosphorylation of VEGFRs is followed by downstream activation of MAPK/ERK, PI3K/Akt, PLC/PKC and other signaling pathways [26,44,74,79,109]. The activation of these pathways, brought by autocrine VEGF signaling and subsequent VEGFR dimerization, has also been observed in a variety of malignancies, promoting survival, proliferation and invasion of cancer cells [44,63,64,66,74,79,86,99]. Hypoxia can further provide these cancer cells with a survival and growth advantage by inducing the expression of VEGFs and VEGFRs [26,45,64,79]. Apart from the classical VEGF receptors, studies on neuropilin have highlighted its role as a critical co-receptor that facilitates VEGF signaling [79,110]. Indeed, NRP1 and NRP2 expression has been observed in cancer cells demonstrating a functional role [63,79,80,110]. Autocrine VEGF/VEGFR signaling was found to be enhanced by interaction with NRP1 in glioblastoma multiforme [54], while in NSCLC NRP1 overexpressing tumor cells exhibited significantly increased tumor growth [63]. In breast cancer, binding of VEGF to neuropilin enhanced cancer cell survival with additional evidence showing that NRP1 supports VEGF autocrine invasive function and chemotaxis of breast cancer cells [79,80,81].

ΕΜΤ, the process by which epithelial cells can acquire mesenchymal features, has emerged as an integral process of cancer progression [111,112]. In addition, endothelial cells undergo a phenotypic switching, known as endothelial-to-mesenchymal cell transition (EndMT), which is essential during angiogenesis [113], with several EMT markers being associated with a pro-angiogenic phenotype [111]. In cancer, the process of EndMT produces cells with fibroblast-like properties which serve as cancer-associated fibroblasts facilitating tumor progression [114,115]. Furthermore, the crosstalk between VEGF and Notch pathways has been established to promote EndMT in endothelial cells of tumors [116], while the addition of VEGF was shown to induce EMT in A549 lung cancer cells [117] and elicit the appearance of EMT markers in pre-invasive prostate cancer cells [118]. These findings show the interdependent nature of angiogenesis, EndMT and EMT in promoting carcinogenesis [111,113]. Likewise, along with inducing tumor growth via an autocrine mechanism, evidence suggests that VEGFR expression in tumor cells also promotes their migration and induces EMT [51,100]. In breast cancer, the expression of EMT markers, including Twist1 and vimentin, was higher in tumors with greater VEGFR2 expression, while E-cadherin expression was lower in the same tumors [100]. Furthermore, VEGF signaling in breast cancer cells was found to promote changes stimulating their invasion [76]. Indeed, VEGF singling induced the expression of the CXCR4 chemokine receptor in breast cancer cells by employing the NRP1 receptor. This demonstrated that the VEGF pathway can direct the migration of cancer cells towards specific chemokines and promote breast carcinoma invasion, while no evidence was shown to suggest that this particular pathway would enhance the survival of these cancer cells [80]. In colorectal carcinoma, VEGF stimulation resulted in enhanced cell migration linked to the activation of focal adhesion components that regulate this process. Cell migration was effectively blocked by pharmacologic inhibition of VEGFR1 or Src kinase, suggesting that VEGFR1 promotes migration of tumor cells through a Src-dependent pathway [66]. In addition, metastatic colon cancer cells were found to be dependent on VEGFR1 signaling for their survival [68,69]. VEGFR1 activation by VEGF-A or VEGF-B was also found to promote migration and invasion of pancreatic carcinoma cell lines without appearing to enhance cancer cell proliferation [61]. Likewise, invasion and metastasis of pancreatic neuroendocrine tumors was suppressed with simultaneous inhibition of c-MET and VEGF signaling [119].

Nonetheless, solid conclusions on the role of the VEGF/VEGFR pathway in promoting autocrine stimulation of tumor cell migration and invasion, are difficult to be drawn, and are perhaps cell- and context-dependent, since contrary to the above, VEGF was demonstrated to negatively regulate tumor cell invasion and mesenchymal cell transition through a MET/VEGFR2 complex in glioblastoma mouse models [120]. Moreover, despite evidence suggesting that NRP1 is implicated in breast carcinoma invasion, NRP1 expression on prostate cells was strongly and negatively correlated with the ability of these cell lines to invade and migrate [95].

## 4. Immunomodulatory Effects of the VEGF/VEGFR Pathway

In addition to its various roles in angiogenesis and direct stimulation of tumor cells’ survival, proliferation and invasion, VEGF can also have immunosuppressive effects [15,121]. Over the last several years, cancer immunotherapy has emerged as a major therapeutic modality, revolutionizing medical oncology [5,13,15]. Its success relies on the recruitment, expansion and effective anticancer activity of immune effector cells within the tumor microenvironment (TME) [13]. Despite rapidly transforming anticancer treatment and providing with durable responses, many patients do not derive benefit from this approach [5,13]. Human cancer cells can employ multiple immune inhibitory mechanisms, resulting to immune escape and likely explaining the lack of response observed in several cases [5]. One such mechanism relates to VEGF, hence combination with anti-angiogenic agents, is one of the many strategies currently under investigation to improve the response rates and duration of immunotherapies [5,15,122,123] (Table 2).

### 4.1. Immune Cell Infiltration

Infiltration of tumors by immune cells is a multistep process involving trafficking of immune cells to the tumor blood vessels, adhesion to the endothelium and ultimately crossing the endothelial barriers into the TME [13,15]. Extravasation into the tumor tissue is dependent upon interactions with adhesion molecules expressed on the immune cells themselves and the luminal surface of the tumors’ endothelial lining such as E-cadherin, intercellular adhesion molecule-1 (ICAM-1) and vascular cell adhesion molecule-1 (VCAM-1) [13,15,121]. VEGF is suggested to impair interactions between leukocytes and endothelial cells by downregulating the expression of these adhesion molecules or inhibiting their clustering [13,121,152,153,154]. Indeed, sunitinib treatment resulted in upregulated expression of ICAM-1 and VCAM-1 adhesion molecules on endothelial cells of tumor bearing mice [141], with several studies reporting that infiltration of TILs is markedly increased in animal tumor models and in humans after VEGF inhibition [5,129,132,141,155]. Furthermore, infiltration of immune cells into the TME is further hindered by the structurally and functionally abnormal tumor vessels [13]. It is suggested that judicious doses of anti-angiogenic agents have the potential to improve the effectiveness of immunotherapy by transiently restoring the abnormal tumor vasculature and thus increasing the infiltration of immune effector cells into the TME [13,14,156,157,158]. Likewise anti-angiogenic agents were found to induce the formation of high endothelial venules (HEVs) that further promote lymphocyte infiltration [159].

### 4.2. Effector T-cells

VEGFR expression has been detected on T-cells, with several reports suggesting that VEGF signaling can directly affect T-cells’ development, homing and cytotoxic functions [15,121]. The activation of the MAPK/ERK and PI3K/Akt pathways after VEGF stimulation on CD4+CD45RO+ memory T-cells that express VEGFR1 and VEGFR2 provides evidence of a functional VEGF/VEGFR interaction [160]. Contrary to its suppressive role, VEGF induced, via VEGFR2, production of pro-inflammatory molecules, such as INF-γ and IL-2 and stimulated migratory responses in these memory CD4+ T-cells [160]. Nonetheless, mounting evidence support the suppressive effects of VEGF on effector T-cells [121,161]. Specifically, Ohm et al. [162] reported that VEGF can impede with the differentiation of hematopoietic progenitor cells in the thymus into CD8+ and CD4+ T-cells [15,162]. Furthermore, CD3+ T-cells’ proliferation and cytotoxic effects were directly suppressed by VEGF upon its binding to VEGFR2 expressed on the activated effector T-cells’ surface [163,164]. VEGF-A also contributes to CD8+ T-cells exhaustion, in a VEGFR2 and NFAT (nuclear factor of activated T-cells) dependent manner, by promoting the expression of checkpoint molecules such as programmed cell death protein 1 (PD-1), cytotoxic T-lymphocyte-associated protein 4 (CTLA-4), T-cell immunoglobulin mucin receptor 3 (TIM3) and lymphocyte activation gene 3 protein (LAG3); thus, resulting in the development of an immunosuppressive microenvironment that could be reverted upon VEGF-A/VEGFR inhibition by anti-angiogenic agents [165]. VEGF can also indirectly suppress effector T-cells functions by inducing Fas-Ligand expression on endothelial cells, resulting in a selective barrier that causes apoptosis of infiltrating CD8+ T-cells, but not of Tregs [166]. Interfering with the VEGF/VEGFR interaction on T-cells has shown promising results in enhancing anti-tumor immunity [121]. Notably, patients with metastatic colorectal cancer displayed increased B- and T-cell compartments after treatment with bevacizumab [126]. Decreased levels of pro-angiogenic mediators and inflammatory cytokines were also observed after addition of bevacizumab to concomitant chemotherapy in patients with NSCLC, resulting to improved DC activation and T-cell cytotoxicity [127]. Likewise, sunitinib, a multi-tyrosine kinase inhibitor, displayed increased Th1 responses by reducing the expression of inhibitory molecules including TGFβ, IL-10, Foxp3, PD-1 and CTLA4 [143].

### 4.3. Regulatory T-cells (Tregs)

Contrary to its inhibitory effects on effector T-cells, VEGF signaling seems to play a role in inducing and/or maintaining Foxp3+ regulatory T-cell populations (Tregs) in patients with cancer [167]. Regulatory T-cells exert immunosuppressive effects on effector T-cells [121,168], with evidence suggesting that VEGF induces Tregs proliferation through VEGFR2 activation [124]. Similarly, interaction of VEGF with NRP1 expressed on Tregs was found critical for tumor homing, since by abolishing NRP1 expression Tregs populations were reduced, resulting in CD8+ T-cells raise in melanoma mouse models [169]. Treatment with anti-angiogenic agents is considered to reverse VEGF induced promotion of Tregs; as expected, bevacizumab demonstrated inhibition of Treg accumulation in peripheral blood of patients with metastatic colorectal cancer [124]. Likewise, a decrease in regulatory T-cell numbers was evident after sunitinib treatment in tumor bearing mice and in patients with metastatic renal cancer [124,136,138,139]. It is therefore reasonable that anti-angiogenic agents are expected to modulate anti-tumor immunity by interfering with inhibitory Tregs [121].

### 4.4. Dendritic Cells (DCs)

One of the first described immunosuppressive functions of VEGF would be hindering dendritic cell (DC) maturation [15,170]. This is evident by the defective or reduced numbers of mature DCs reported in several malignancies to be inversely corelated with VEGF plasma concentrations [171,172,173,174]. Inhibition of NF-κB signaling is suggested to be the underlying mechanism that impairs DCs’ differentiation and maturation, with various studies indicating this to be a direct consequence of VEGF binding to either VEGFR2 or VEGFR1 on DCs [15,170,175,176,177]; although NRP1 is also implicated [178], as well as PLGF binding to VEGFR1 [177,179]. Along with directly affecting DCs’ maturation, VEGF was also found to upregulate PDL1 on DCs, resulting in inhibition of T-cells’ expansion and function [180]. DCs are antigen-presenting cells, integral for a successful immune response, and thus targeting factors, such as VEGF, that interfere with DCs’ differentiation, maturation and activation is a reasonable therapeutic strategy. Bevacizumab has shown promising results in reversing the VEGF-induced inhibition of differentiation of monocytes into DCs in vitro [131], as well as in restoring peripheral blood DC numbers in cancer patients and promoting their activation [127,130]. Sorafenib and sunitinib, two multi-kinase inhibitors, have also shown effects on DCs, although discrepancies lie among different studies making their exact role debatable and perhaps context-dependent [121,131,151].

### 4.5. Myeloid Derived Suppressor Cells (MDSCs)

Myeloid derived suppressor cells (MDSCs) form a heterogenous group of myeloid origin immune cells, that are frequently present in pathologic conditions characterized by chronic inflammation. Increased intratumoral VEGF concentration has been corelated with the presence of MDSCs [181], with several studies indicating that VEGF can promote the accumulation of MDSCs in tumors and peripheral blood of cancer patients, via VEGFR2-STAT3 activation, but not VEGFR1 [139,181,182]. MDSCs are known for their immunosuppressive properties [183] that stem from their ability to inhibit T-cell proliferation and activation, and when activated by VEGF, MDSCs could also stimulate the development of other immunosuppressive cells including Tregs [121,184,185,186]. Angiogenetic agents like sunitinib, axitinib, sorafenib and bevacizumab have demonstrated ability to constrain the MDSC compartment and reduce their suppressive capacity resulting in a more favorable microenvironment [125,139,140,143,148,187].

### 4.6. Tumor Associated Macrophages (TAMs)

Macrophages are important cells of the innate immunity and play a central role in inflammation [188]; however, macrophages that are present in the TME in high numbers, also known as tumor associated macrophages (TAMs), are suggested to display a tumor promoting phenotype [189,190]. VEGF and most likely PLGF are reported to act as chemoattractants for monocytes via activation of VEGFR1 [15,191]. VEGF-A could thus recruit macrophages to tumors with high VEGF expression and contribute to tumor growth by establishing an immunosuppressive microenvironment [190,192]. In addition to their immunosuppressive functions TAMs are implicated in the development of resistance to anti-VEGF agents [193]. Reducing the recruitment of TAMs or reprogramming M2-like TAMs towards an anticancer M1 phenotype [13] seems a reasonable strategy to reverse immunosuppression, as well as to deal with anti-VEGF resistance, especially in glioblastoma were increased TAMs have been correlated with poor prognosis and disease progress on bevacizumab [194,195,196].

### 4.7. Combinations of VEGF/VEGFR Inhibition with Cancer Immunotherapy

Cumulative evidence provides the rationale that anti-angiogenic treatment might augment the efficacy of immunotherapy and several recent pre-clinical models and clinical studies have tested this hypothesis. In a pre-clinical study, sunitinib was reported to exert potent complementary anti-tumor effects when combined with CD40-stimulating immunotherapy, by mediating DCs activation, reducing MDSCs and increasing endothelial activation that resulted in enhanced recruitment of cytotoxic T-cells [141]. Dual VEGF-A and Ang2 inhibition displayed enhanced anti-tumor immunity with PD1 blockade in breast, melanoma and pancreatic neuroendocrine tumor models [156]. Likewise, simultaneous blockade of PD-1 and VEGFR2, in a Colon-26 adenocarcinoma mouse model, induced a synergistic in vivo anti-tumor effect [197]. In a mouse model of SCLC, combined treatment with anti-VEGF and anti-PDL1 targeted therapy provided improved treatment outcome compared with anti-PDL1 or anti-VEGF monotherapy [198]. Co-administration of low-dose apatinib, a VEGFR2-TKI, with PDL1 inhibition resulted in reduced tumor growth, fewer metastases and prolonged survival of lung cancer mouse models [146]. Alleviated hypoxia, increased infiltration of CD8+ T-cells, reduced recruitment of TAMs and decreased TGFβ was observed with low-dose apatinib [146]. Anticancer activity of combining apatinib with anti-PD1 was also evident in a small cohort of pretreated patients with advanced NSCLC [146]. The treatment effect of axitinib, a TKI against VEGFR1/2/3, combined with CTLA4 blockade was investigated in a mouse melanoma model. Combination of anti-angiogenesis and checkpoint inhibition resulted to an increased anti-tumor effect and survival, partially due to enhanced immune response generated by an increased antigen-presenting function of intratumoral DCs in combination with a reduced suppressive capacity of intratumoral MDSCs [145].

Following the observation that metastatic melanoma patients with high levels of VEGF presented worse survival when treated with ipilimumab, a CTLA4 inhibitor [199], a phase I trial was conducted to investigate the combination of ipilimumab with bevacizumab. The trial demonstrated that VEGF-A blockade influences inflammation, lymphocyte trafficking and immune regulation, and was associated with favorable clinical outcome in metastatic melanoma patients [128,200]. Further analysis showed that the combination therapy elicited humoral immune responses against galectin-1, which exhibited protumor, pro-angiogenesis and immunosuppressive activities in 37.2% of treated patients [200]. The first ever phase III trial to successfully investigate the synergistic effect of immune-checkpoint inhibition with VEGF blockade was the Impower150 in NSCLC [201]. This pivotal study demonstrated that the addition of atezolizumab to bevacizumab plus chemotherapy significantly improved PFS and OS among patients with metastatic non-squamous NSCLC, regardless of PD-L1 expression and EGFR or ALK genetic alteration status [201]. Of note, while the quadruplet combination (i.e., atezolizumab, carboplatin, paclitaxel, bevacizumab) was superior to chemotherapy plus bevacizumab, the atezolizumab plus chemotherapy combination was not, thus supporting the modulatory role of anti-angiogenesis to immunotherapy. In metastatic renal cell carcinoma (mRCC) phase I studies combining a VEGF-TKI and PDL1/PD1 blockade, suggest that anti-angiogenesis could potentiate PDL1/PD1 inhibition. Specifically, tissues from patients with mRCC exhibited increased intra-tumoral CD8+ T-cells after combination treatment with bevacizumab and atezolizumab, a PDL1 inhibitor [129], while co-administration of axitinib plus pembrolizumab or avelumab, showed promising anti-tumor activity in patients with treatment-naive advanced RCC in phase I trials [202,203]. A similar but somewhat distinct therapeutic approach in a phase II trial combining dendritic cell-based immunotherapy with sunitinib, also demonstrated benefit for patients with mRCC [204]. Another phase II trial in RCC that compared atezolizumab plus bevacizumab against sunitinib, displayed enhanced efficacy for the combination in PDL1-positive patients, while sunitinib monotherapy had better results in in patients with predominant angiogenesis markers [205]. The phase III trial that followed, IMmotion151, confirmed prolonged PFS for the atezolizumab plus bevacizumab in the PDL1 positive population, however, longer follow-up is warranted to establish whether a survival benefit will emerge [206]. Likewise, two other phase III clinical trials have investigated the combination of an immune-checkpoint inhibitor with axitinib, in untreated patients with mRCC [207,208]. The JAVELIN Renal 101 trial reported a significantly longer progression free survival (PFS) for the combination of axitinib with avelumab, a PDL1 inhibitor, against sunitinib [207], while the KEYNOTE-426 trial resulted in both overall survival and PFS benefit for the combination of pembrolizumab with axitinib compared to sunitinib, regardless of PDL1 expression [208]. Both trials reported increased objective response rates for the combination therapy. VEGF inhibition and PDL1 blockade has also led to promising results in hepatocellular carcinoma. Specifically, a phase 1 b trial in patients with unrespectable hepatocellular carcinoma the combination of atezolizumab plus bevacizumab reported PFS benefit [209], leading to a phase III trial against sorafenib, where once again the combination treatment was superior in terms of OS and PFS [210] (Table 3).

## 5. Conclusions and Future Directions

Until recently, anti-angiogenic factors were considered to exert their anti-tumor effects by inhibiting the formation of new blood vessels; however, growing evidence suggests that inhibition of the VEGF/VEGFR pathway may have multiple effects including direct inhibition of tumor cell growth and immunostimulatory functions [15,16,54,58].

VEGF-mediated autocrine-paracrine loops that directly influence and promote tumor cell survival, proliferation and invasion have been identified in several cancers including lung, breast, prostate, bladder, colorectal, pancreatic, sarcomas, ovarian, melanoma, gliomas and hematopoietic malignancies. Moreover, this autocrine or paracrine loop represents an attractive therapeutic target [13]. Indeed, it has been shown that a natural occurring, soluble form of NRP1 can act as a VEGF_165_ antagonist exhibiting anti-tumor activity in vivo [211]. Furthermore, in patients with inflammatory, locally advanced breast cancer, bevacizumab was reported to induce apoptosis in tumor cells, along with its inhibitory effects on VEGFR2 activation and permeability [212]. In gliomas, high doses of bevacizumab were suggested to have anticancer properties in vivo, not related to angiogenesis, as regression of glioma cells was demonstrated to occur independently from vascular regression [213]. Likewise, anti-VEGF treatment was explored in multiple myeloma (MM) cells. VEGF-A blockade caused cytostasis in MM cells, demonstrating that bevacizumab has a direct influence on major pathways critically activated in MM that is independent from its established effect on angiogenesis [92]. In a preclinical study, chronic exposure of colorectal cells to bevacizumab upregulated VEGF-A, -B, -C, PLGF, VEGFR1 expression and VEGFR1 phosphorylation, resulting to increased tumor cell migration and invasion and enhanced metastatic potential [214], highlighting the rationale of successfully blocking the VEGF/VEGFR pathway directly on tumor cells.

The immunosuppressive properties of VEGF likely stem from its role in initiating the wound healing process, which benefits from down-modulating cellular immunity and stimulating angiogenesis and tissue growth and repair [53,215]. Accordingly, the immunostimulatory effects of anti-angiogenetic agents can be condensed to at least four different functions: (a) preventing the VEGF-mediated inhibition of effector T-cells trafficking, proliferation and cytotoxic functions, thus enhancing T-cell mediated immune response; (b) restoring DCs’ differentiation and maturation, thus promoting antigen presentation and T-cell activation; (c) hindering the recruitment of inhibitory cells in the TME, such as Tregs, MDSCs and M2-like TAMs; and (d) activating the tumor endothelium and inducing normalization of the disorganized, leaky and abnormal tumor vasculature that results to hypoxia, hinders effector T-cell infiltration and fosters immunosuppression in the TME [13,53,158,165,170,176,189,216]. Bevacizumab, in particular, has been found to relieve immunosuppression by decreasing MDSCs and Tregs populations, and also to enhance cytotoxic T-lymphocytes responses, improve DCs maturation and increase T-cells infiltration [124,125,127,128,130,132,133,134,135]. An inhibitory effect on MDSCs and/or Tregs compartments has also been displayed for sorafenib [148,150], axitinib [144,145] and sunitinib [136,137,138,139,140]. Likewise, a positive reinforcement on T-cells recruitment and functions has been observed for sunitinib [141,142], axitinib [144], apatinib [147] and sorafenib [150]. Another important function of sunitinib and apatinib is their ability to decrease the expression of inhibitory checkpoint molecules [141,143,147]. Furthermore, DCs’ functions were enhanced with axitinib [145], while sorafenib provided contradictory results regarding DCs and T-cells regulation [131,151] (see Table 2).

Taking advantage of their immunomodulatory functions, several clinical trials in melanoma, RCC, NSCLC and hepatocellular carcinoma have successfully evaluated the combination of checkpoint inhibition with VEGF/VEGFR blockade, providing evidence of the efficacy of such an approach. Ongoing phase I to III clinical trials continue to explore the efficacy of a combinational strategy in a variety of different malignancies, including gynecological, gastrointestinal, genitourinary, central nervous system (CNS), lung and several others advanced solid tumors. Phase III clinical trials of an immune-checkpoint inhibitor with an anti-angiogenic agent combination that are currently active can be reviewed in Table 3. Ongoing research has the dynamic to expand the therapeutic indications of this strategy. However, despite, the synergetic therapeutic effect displayed so far in many trials, not all them reported positive outcomes; notably, in a phase I trial, the combination of tremelimumab, an anti-CTLA4 antibody with sunitinib induced severe toxicities, including kidney failure, in patients with mRCC [137], highlighting the importance of carefully designed clinical trials that will allow us to minimize toxicities and safely evaluate these combinations [13].

It is well accepted by now that not all patients will achieve benefit from immune-checkpoint inhibition, hence the need for identifying novel approaches [5]; at the same time, previously unappreciated studies on VEGF biology have demonstrated immunomodulatory properties and tumor regression by disrupting the VEGF/VEGFR pathway and now provide the scientific basis for new combinational treatments. To date therapeutic indications of anti-PD1/PDL1 combinations with anti-angiogenetic agents include RCC (pembrolizumab or avelumab plus axitinib), NSCLC (atezolizumab plus bevacizumab plus chemotherapy), hepatocellular carcinoma (atezolizumab plus bevacizumab). Ongoing and future studies will likely expand these therapeutic indications, while investigators have also the obligation to identify those patients that would safely benefit from such an approach.

## Figures and Tables

**Figure 1 cancers-12-03145-f001:**
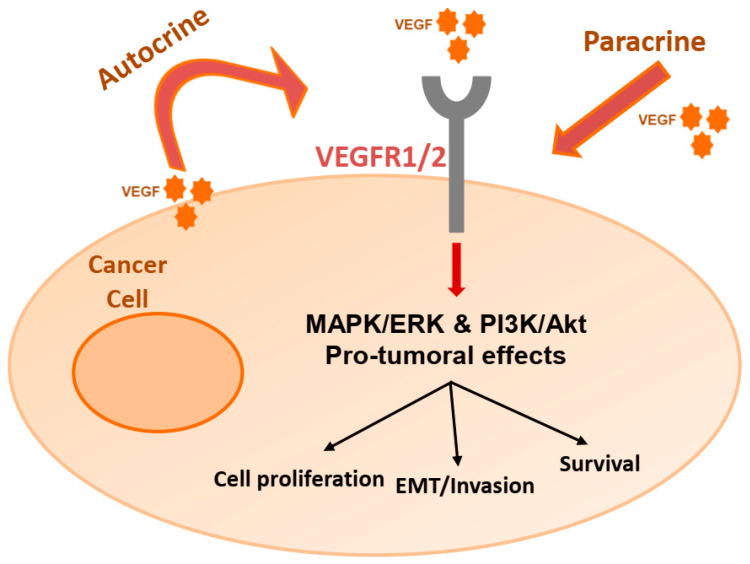
Non-angiogenic effects of the VEGF–VEGFR interaction in cancer cells. VEGF-mediated autocrine-paracrine loops directly influence and promote tumor cell survival, proliferation and invasion. VEGF(R): vascular endothelial growth factor (receptor); EMT: epithelial-mesenchymal transition; MAPK/ERK: Mitogen-activated protein kinase/extracellular signal-regulated kinase; PI3K/Akt: Phosphoinositide 3-kinase/Akt

**Table 1 cancers-12-03145-t001:** Effects of VEGF/VEGFR autocrine-paracrine signaling in cancer cells.

Cancer Type	Effects
Melanoma (VEGFR1/2; NRP1/2)	Enhances the proliferation of melanoma cells [59]. Mitigates melanoma cells migration (through a NRP1/VEGFR2-mediated response) [60].
Pancreatic (VEGFR1/2; NRP1)	Was shown to activate the MAPK/ERK pathway [44,61]. Stimulates cancer cell growth [44]. Promotes cancer cell migration and invasion, without affecting proliferation (VEGFR1-mediated effect) [61]. Promotes pancreatic cancer aggressiveness by TGFβ1-induced fibrosis and endothelial-to-mesenchymal transition (NRP1-mediated effect) [62].
NSCLC (VEGFR1/2; NRP1/2)	Induces PI3K/Akt and MAPK/ERK activation [63]. Stimulates tumor growth and proliferation of NRP1-expressing cells (VEGFR2/NRP1-mediated effect) [63].
SCLC (VEGFR2/3)	Promotes VEGFR2/3 activation resulting MAPK/ERK phosphorylation [64]. Induces cancer cell proliferation [64].
Colorectal (VEGFR1)	Promotes Akt and ERK phosphorylation [65]. Enhances survival and resistance to chemotherapy of cancer cells [65]. Was shown to enhance cellular migration and promote tumor progression and metastasis [66,67]. Was found to support the survival of cancer cells undergoing EMT [68,69].
Gastric (VEGFR1/2)	Stimulates tumor growth (VEGFR2-mediated response) [57,70].
Prostate (VEGFR1/2)	Was shown to enhance prostate cancer cells proliferation (VEGFR2-mediated effect) [71,72].
Glioblastoma (VEGFR1/2; NRP1)	Promotes MAPK/ERK, PI3K/Akt and PLC/PKC pathways activation [73,74]. Stimulates proliferation of glioma cells (VEGFR2-mediated response) [75]. Supports tumor growth (VEGFR1/2-mediated effect) [73].
Breast cancer (VEGFR1/2; NRP1)	Induces activation of the MAPK/ERK and PI3K/Akt pathways [76]. Supports tumor cells survival, stimulates their proliferation and contributes to mammary tumor growth [77,78,79,80,81]. Induces invasion and chemotaxis of breast cancer cells and enhances EMT [79,80,81,82]. Inhibits apoptosis and protects from chemotherapy [79,83]. Confers cancer stem cells traits in breast cancer cells and was found to drive cancer stem cells self-renewal [84,85].
Head & Neck (VEGFR2)	Regulates proliferation and invasion of head & neck cancer cells [86].
Bladder (VEGFR1/2)	Enhances survival and proliferation of bladder cancer cells (VEGFR2-mediated effect) [45,87].
Rhabdomyosarcoma (VEGFR1/2)	Increases cancer cell proliferation (VEGFR1-mediated effect) [52].
Ovarian (VEGFR2)	VEGFR2-phosphorylation has been corelated with ovarian cancer cell survival and proliferation [58].
Multiple Myeloma (VEGFR1)	Mediates activation of the MAPK/ERK, PI3 k/PKC and McL1/survivin pathways resulting in increased proliferation, migration and survival [88,89,90,91,92].

Abbreviations: EMT: epithelial–mesenchymal transition; VEGF(R): vascular endothelial growth factor (receptor); NRP: neuropilin; MAPK: Mitogen-activated protein kinase; ERK: extracellular signal-regulated kinase; TGFβ1: Transforming growth factor beta 1; PI3K: Phosphoinositide 3-kinase; Akt: Protein kinase B; PLC:phospholipase C; PKC: Protein kinase C; McL1: myeloid cell leukemia 1.

**Table 2 cancers-12-03145-t002:** Immunomodulatory effects of selected anti-angiogenic factors.

Anti-Angiogenic Agent		Functions
VEGF-A antibody	Bevacizumab	Decreases MDSCs and Tregs accumulation [124,125].
		Enhances CTLs responses: It was shown to (a) increases the peripheral B- and T-cell compartments [126], (b) correlate with an increase in activated (CD8+ CD62 L+) CTLs, long-term effector memory (CD8+ CD27+) and central-memory (CD8+ C45 RA-CCR7+) CTLs [127,128] and (c) enhance antigen-specific T-cell migration [129]
		Improves DCs maturation and activation: It was shown to increase the percentage of activated and mature myeloid derived DC [127,130], and to reverse the VEGF inhibitory effects on DCs [131].
		Induces vessel normalization, increases tumor vascular expression of ICAM1 and VCAM1 and T-cell tumor infiltration [132,133,134,135].
VEGFR1–3, PDGFR, c-KIT, FLT-3, CSF-1 R and RET mtTKI	Sunitinib	Enhances the Th1 immune response and inhibits the immunosuppressive Th2 response [136,137].
		Decreases MDSCs and tumor Tregs compartments [136,137,138,139,140].
		Induces endothelial activation and T-cell recruitment, by enhancing the expression of chemokines and adhesion molecules on tumor endothelial cells, resulting in a higher number of CD3+ T-cells in the tumor [141,142].
		Enhances the percentage and number of intratumoral CD4 and CD8 T-cells and decreases the expression of inhibitory molecules (i.e., CTLA-4 and PD-1) on TILs [141,143].
VEGFR1–3, PDGFR and c-KIT mtTKI	Axitinib	Enhances the CD8+ T cells compartment [144].
		Increases the antigen-presenting function of intratumoral DCs [145].
		Reduces MDSCs levels [144] and inhibits their suppressive capacity [145].
VEFGR2 TKI	Apatinib	Increases the infiltration of CD8+ T cells and reduces the recruitment of TAMs [146].
		Reduces the expression levels of inhibitory checkpoint molecules, such as Lag-3, PD-1 and Tim3 in CD8+ T cells [147].
		Enhances the production of IFN-γ and IL-2 and promote the cytotoxicity of T cells [147].
Raf, VEGFR2, PDGFR, FLT3, RET and c-KIT mtTKI	Sorafenib	Reverses immunosuppression: It decreases MDSCs levels [148], Tregs and Th2-cells [149], and inhibits Tregs functions [150].
		Upregulates tumor-specific effector T-cells functions [150] and induces Th1 dominance [149].
		Reverses the VEGF inhibitory effects on DCs [131], but was also shown to inhibit the function of DCs [151] and inhibit the induction of antigen-specific T cells [151].

Abbreviations: (mt)TKI: (multi-target) tyrosine kinase inhibitor; PDGFR: platelet derived growth factor receptor; VEGF(R): vascular endothelial growth factor (receptor); FLT3: Fms-like tyrosine kinase-3; CSF-1 R: colony stimulating factor receptor; RET: glial cell-line derived neurotrophic factor receptor; DCs: dendritic cells; ICAM1: intercellular adhesion molecule-1; VCAM1: vascular cell adhesion molecule-1; Th(1/2): T helper cell (1/2); Lag3: lymphocyte activation gene 3 protein; Tim3: T-cell immunoglobulin mucin receptor 3; PD-1: programmed cell death protein 1; CTLs: cytotoxic T-lymphocytes; Tregs: T-regulatory cells; MDSCs: myeloid derived suppressor cells; TAMs; tumor associated macrophages.

**Table 3 cancers-12-03145-t003:** Phase III studies of immune-checkpoint inhibitors with anti-angiogenic agents.

Cancer Type	Immunotherapy	Anti-Angiogenic Agent	Indication	Year	Current Status	Identifier
Gastrointestinal	Atezolizumab (Anti-PDL1)	Bevacizumab	dMMR, Metastatic CRC	2016	Suspended	NCT02997228
	Nivolumab (Anti-PD1)	Bevacizumab	Metastatic CRC, 1st line	2018	Active, not recruiting	NCT03414983
	Sintilimab (Anti-PD1)	Bevacizumab	RAS-Mutant, Metastatic CRC, 1st line	2019	Not yet recruiting	NCT04194359
	HLX10 (Anti-PD1)	HLX04 (Anti-VEGF)	Metastatic CRC, 1st line	2020	Not yet recruiting	NCT04547166
	Atezolizumab (Anti-PDL1)	Bevacizumab	Advanced HCC, 1st line	2018	Active, not recruiting	NCT03434379
	HLX10 (Anti-PD1)	HLX04 (Anti-VEGF)	Advanced or Metastatic HCC, 1st line	2020	Not yet recruiting	NCT04465734
Genitourinary	Pembrolizumab (Anti-PD1)	Axitinib	Untreated, advanced RCC	2016	Active, not recruiting	NCT02853331
	Pembrolizumab (Anti-PD1)	Lenvatinib	Untreated, advanced RCC	2016	Active, not recruiting	NCT02811861
	Atezolizumab (Anti-PDL1)	Bevacizumab	Untreated, advanced RCC	2015	Active, not recruiting	NCT02420821
	Avelumab (Anti-PDL1)	Axitinib	Untreated, advanced RCC	2016	Active, not recruiting	NCT02684006
	Nivolumab (Anti-PD1)	Cabozantinib	Untreated, metastatic RCC	2019	Recruiting	NCT03793166
	Anlotinib (anti-PDL1)	TQB2450 (mtTKI)	Advanced RCC	2020	Recruiting	NCT04523272
	Toripalimab (anti-PD1)	Axitinib	Unresectable or Metastatic RCC, 1st line	2020	Recruiting	NCT04394975
Lung	Atezolizumab (Anti-PDL1)	Bevacizumab	Stage IV Non-Squamous NSCLC, 1st line	2015	Active, not recruiting	NCT02366143
	Atezolizumab (Anti-PDL1)	Bevacizumab	Stage IV Non-Squamous NSCLC, 1st line	2019	Recruiting	NCT04194203
	Sintilimab (Anti-PD1)	IBI305 (Anti-VEGF)	EGFR-mutated, TKI-resistant, Locally Advanced or Metastatic, non-squamous NSCLC	2019	Recruiting	NCT03802240
	HLX10 (Anti-PD1)	HLX04 (Anti-VEGF)	Stage IIIB/IIIC or IV non-squamous NSCLC	2019	Recruiting	NCT03952403
Gynecological	Atezolizumab (Anti-PDL1)	Bevacizumab	Platinum-Resistant, Recurrent, Ovarian, Fallopian Tube, or Peritoneal Cancer	2016	Recruiting	NCT02839707
	Atezolizumab (Anti-PDL1)	Bevacizumab	Platinum-Sensitive Relapse, Ovarian, Fallopian Tube, or Peritoneal Cancer	2016	Active, not recruiting	NCT02891824
	Atezolizumab (Anti-PDL1)	Bevacizumab	Stage III/IV Ovarian, Fallopian Tube, or Peritoneal Cancer	2017	Active, not recruiting	NCT03038100
	Atezolizumab (Anti-PDL1)	Bevacizumab	Persistent, Recurrent or Metastatic (Stage IVB) Cervical Cancer	2018	Recruiting	NCT03556839
	Pembrolizumab (Anti-PD1)	Bevacizumab	Persistent, Recurrent or Metastatic Cervical Cancer	2018	Active, not recruiting	NCT03635567
	Dostarlimab (Anti-PD1)	Bevacizumab	Stage III/IV Nonmucinous Ovarian Cancer, 1st line	2018	Recruiting	NCT03602859
	BCD-100 (Anti-PD1)	Bevacizumab	Advanced Cervical Cancer, 1st line	2019	Recruiting	NCT03912415

Abbreviations: PD(L)1: programmed death (ligand) 1; VEGF: vascular endothelial growth factor; mtTKI: multi-target tyrosine kinase inhibitor; HCC: hepatocellular carcinoma; NSCLC: non-small cell lung cancer; RCC: renal cell carcinoma; CRC: colorectal cancer; dMMR: deficient mismatch repair; EGFR: epidermal growth factor receptor.

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
