# Peer review of "Old Player-New Tricks: Non Angiogenic Effects of the VEGF/VEGFR Pathway in Cancer"

_cancers, 2020, doi:10.3390/cancers12113145_

Round 1
Reviewer 1 Report
In the following review, the authors analyze the angiogenesis pathway in tumors and in particular how anti-angiogenic agents that interrupt the VEGF / VEGFR pathway play a considerable role. Research on VEGF-mediated signaling originally focused on its survival effect and mitogenic effects on endothelial cells, not finding the desired success in anti-angiogenic therapy. on the other hand, VEGF can have multiple effects on other types of cells, including immune and tumor cells, directly influencing and promoting their survival and proliferation as well as invasion and contributing to an immunosuppressive microenvironment. In this review, the authors focus attention on the effects of the VEGF / VEGFR pathway in non-endothelial cells, as well as anti-angiogenic agents which include direct inhibition of tumor cell growth and immunostimulatory function. Last but not least, they highlight how the interruption of the VEGF / VEGFR pathway is today the basis of immunotherapy treatments with anti-angiogenic agents.
Given the topic dealt with by the authors, being strongly correlated to the epithelium / mesenchyme transition that occurs during tumor transformation, it would have been appropriate to mention at least in the introductory phase the role played by citing works such as:
- Ribatti D. "Epithelial-mesenchymal transition in morphogenesis, cancer progression and angiogenesis" Exp Cell Res 2017 Apr 1; ​​353 (1): 1-5. doi: 10.1016 / j.yexcr.2017.02.041.
- Ghersi, G. "Roles of molecules involved in epithelial / mesenchymal transition during angiogenesis" Front Biosci 2008 Jan 1; 13: 2335-55. doi: 10.2741 / 2848.
Author Response
We would like to thank the reviewer for the suggestion. The role of EMT and EndMT along with the proposed references and additional references have been added between lines 322-332.

Reviewer 2 Report
Ntellas et al. present an interesting revision on the role of VEGF/VEGFR pathway on additional cell types including immune and tumor cells, by stimulating tumor cell survival, proliferation and invasion and contributing to an immunosuppressive microenvironment. Generally, these additional effects take a second place focusing the attention on the main role of VEGF in promoting angiogenesis. Thus the topic is interesting and suitable.
Minor changes are required:
The role of NOTCH signaling pathway in the modulation of VEGF receptor expression should be examined.
The authors should describe non angiogenic effects of the VEGF/VEGFR pathway not only in solid tumors, but also in hematologic malignancies such as multiple myeloma.
Recently, the adhesion molecule JAM-A, previously described as regulator of angiogenesis and tumor progression in solid tumors, has been described to stimulate angiogenesis also in multiple myeloma and its expression correlate with poor prognosis in relapsed refractory multiple myeloma patients. The authors could provide a little more consideration about this.
A figure that summarises VEGF-VEGFR interaction and the consequent non-angiogenic effects should be included.
The structure of table 2 and 3 is confused and should be revised. Regarding table 2, I suggest to use three columns: the first column to indicate the involved receptor; the second column to show the drug which targets that receptor; and the third column to describe the effects.
In table 3 the authors should always indicate the specific target of the monoclonal antibody (anti-PD1 or anti-PDL1).
The most of references are too old. The authors should cite in the review papers published during last 10 years.
Careful English revision is required.
Author Response
We would like to thank the reviewer for the constructive suggestions.
The role of Notch signaling in the modulation of VEGF receptor expression has been addressed. Please see lines 78-79, 125-134 together with the appropriate references.
The non-angiogenic effects of the VEGF/VEGFR pathway in hematologic malignancies, including multiple myeloma and JAM-A, have also been added, Please see lines 271-288.
Tables 2 and 3 have been reconfigured according to your suggestions.
A figure that summarizes VEGF-VEGFR interaction has been added, please see lines 151-156.
Many references are indeed old, however reflecting that the VEGF/VEGFR pathway is not a new concept in cancer. While more recent references were used for more recent topics, such as the combination of anti-angiogenics with immunotherapy, we felt that we couldn’t exclude older, but pivotal studies on the VEGF/VEGFR pathway.

Reviewer 3 Report
Ntellas and colleagues reviewed non-angiogenic effects of the VEGF/VEGFR pathway in cancer. They focus on the effects on cancer cells and on immune modulation. Overall the manuscript is well written; a number of points should be addressed.
The basis for new combinational treatments of immunotherapy with anti-angiogenic agents is well developed in the manuscript. The part on different cancer cell lines need more precision and critical discussion i.e. the authors correctly describe that “However, anti-angiogenic therapy in general has managed to offer only modest survival benefits before resistance develops, and bevacizumab’s benefit was only evident when combined with cytotoxic chemotherapy [13]”. How this is in line with the pro-tumorigenic effects of VEGF in most cell lines? If the different cell culture models would be relevant for the in vivo situation, one might expect inhibition of tumour cell proliferation and migration in response to bevacizumab?
Specific points:
“with 1 to 5 people losing their life to cancer in the US” is unclear. Please re-phrase.
“Amongst the most important milestones of the 21st century in cancer treatment would be the introduction of anti-angiogenic therapy in 2004 and the emergence of immune checkpoint inhibitors in 2011” is a highly subjective statement.
In the section on VEGF splicing, a and b splice variants should be included.
Line 166: NSCLC should not be confounded with some cancer cell lines.
The cited observations in general should mention the specific cell lines used.
Glioblastoma chapter: Please clarify the cell lines and identify whether it is glioma or glioblastoma.
Author Response
We would like to thank the reviewer for the constructive comments.
The presence of autocrine growth pathways in some tumors implies that VEGF antisense therapy is acting on 2 levels: Antiangiogenic effects on the tumor vasculature and antineoplastic effects on the tumor cell population. We would like to thank the reviewer once again for their suggestion and we now elaborate in the text that the effect on cancer cells would require further investigation for its clinical relevance, please see lines 145-148.
According to the reviewer’s suggestions we have addressed the specific points mentioned.
We have re-edited the section on VEGF splicing to include a and b splicing, please see lines 117-123.
We have also added the specific cell lines used throughout the text.
The glioblastoma chapter has been re-edited, please see lines 226-240.
All phrases the reviewer has pointed out have been re-edited.

Round 2
Reviewer 3 Report
I have no further suggestions. The authors answered to the questions.